# Cancer Histology and Natural History of Patients with Lung Cancer and Venous Thromboembolism

**DOI:** 10.3390/cancers14174127

**Published:** 2022-08-26

**Authors:** Pedro Ruiz-Artacho, Ramón Lecumberri, Javier Trujillo-Santos, Carme Font, Juan J. López-Núñez, María Luisa Peris, Carmen Díaz Pedroche, José Luis Lobo, Luciano López Jiménez, Raquel López Reyes, Luis Jara Palomares, José María Pedrajas, Isabelle Mahé, Manuel Monreal

**Affiliations:** 1Department of Internal Medicine, Clínica Universidad de Navarra, 28027 Madrid, Spain; 2Interdisciplinar Teragnosis and Radiosomics Research Group (INTRA-Madrid), Universidad de Navarra, 28027 Madrid, Spain; 3CIBER Enfermedades Respiratorias (CIBERES), 28029 Madrid, Spain; 4Hematology Service, Clínica Universidad de Navarra, 31008 Pamplona, Spain; 5CIBERCV, Instituto de Salud Carlos III, 28220 Madrid, Spain; 6Department of Internal Medicine, Hospital General Universitario Santa Lucía, Universidad Católica de Murcia, 30107 Murcia, Spain; 7Department of Medical Oncology, Hospital Clínic, 08036 Barcelona, Spain; 8Department of Internal Medicine, Hospital Germans Trias i Pujol, 08916 Badalona, Spain; 9Department of Medicine, Universitat Autònoma de Barcelona, 08193 Barcelona, Spain; 10Institut de Recerca Germans Trias i Pujol, 08916 Badalona, Spain; 11Department of Internal Medicine, Consorcio Hospitalario Provincial de Castellón, 12002 Castellon, Spain; 12Medicine Department, CEU Cardenal Herrera University, 46115 Valencia, Spain; 13Department of Internal Medicine, Hospital Universitario 12 de Octubre, 28041 Madrid, Spain; 14Department of Pneumonology, Hospital Universitario Araba, 01009 Vitoria-Gasteiz, Spain; 15Department of Internal Medicine, Hospital Universitario Reina Sofía, 14004 Córdoba, Spain; 16Department of Pneumonology, Hospital Universitari i Politècnic La Fe, 46026 Valencia, Spain; 17Department of Pneumonology, Hospital Universitario Virgen del Rocío, 41013 Seville, Spain; 18Department of Internal Medicine, Hospital Clínico San Carlos, 28040 Madrid, Spain; 19Department of Internal Medicine, Hôpital Louis Mourier, Colombes (APHP), University Paris 7, 75013 Paris, France; 20Chair for the Study of Thromboembolic Disease, Faculty of Health Sciences, UCAM–Universidad Católica San Antonio de Murcia, 30107 Murcia, Spain

**Keywords:** venous thrombolism, lung cancer, adenocarcinoma lung cancer, cancer associated thrombosis, histology

## Abstract

**Simple Summary:**

Cancer is a widely heterogeneous disease, and the natural history of patients with cancer-associated thrombosis may differ according to the cancer site. Lung cancer is the most common malignancy, and a leading cause of death. A number of studies in the literature suggest that patients with adenocarcinoma may have a worse outcome than those with squamous or other types of lung cancer. The aim of the current study was to assess the potential impact of lung cancer histology on the incidence rates of VTE recurrences, major bleeding, or death appearing during the course of anticoagulation, in patients with lung cancer and VTE. Our findings, obtained from a large series of consecutive patients with lung cancer and VTE (482 patients), reveal important differences between patients with adenocarcinoma vs. other histologies in their outcomes during anticoagulation. This might likely help to design better therapeutic strategies for patients with lung cancer.

**Abstract:**

***Background:*** In patients with lung cancer and venous thromboembolism (VTE), the influence of cancer histology on outcome has not been consistently evaluated. ***Methods:*** We used the RIETE registry (Registro Informatizado Enfermedad TromboEmbólica) to compare the clinical characteristics and outcomes during anticoagulation in patients with lung cancer and VTE, according to the histology of lung cancer. ***Results*:** As of April 2022, there were 482 patients with lung cancer and VTE: adenocarcinoma 293 (61%), squamous 98 (20%), small-cell 44 (9.1%), other 47 (9.8%). The index VTE was diagnosed later in patients with squamous cancer than in those with adenocarcinoma (median, 5 vs. 2 months). In 50% of patients with adenocarcinoma, the VTE appeared within the first 90 days since cancer diagnosis. During anticoagulation (median 106 days, IQR: 45–214), 14 patients developed VTE recurrences, 15 suffered major bleeding, and 218 died: fatal pulmonary embolism 10, fatal bleeding 2. The rate of VTE recurrences was higher than the rate of major bleeding in patients with adenocarcinoma (11 vs. 6 events), and lower in those with other cancer types (3 vs. 9 events). On multivariable analysis, patients with adenocarcinoma had a non-significantly higher risk for VTE recurrences (hazard ratio [HR]: 3.79; 95%CI: 0.76–18.8), a lower risk of major bleeding (HR: 0.29; 95%CI: 0.09–0.95), and a similar risk of mortality (HR: 1.02; 95%CI: 0.76–1.36) than patients with other types of lung cancer. ***Conclusions*:** In patients with lung adenocarcinoma, the rate of VTE recurrences outweighed the rate of major bleeding. In patients with other lung cancers, it was the opposite.

## 1. Introduction

Patients with cancer are at increased risk of venous thromboembolism (VTE), and VTE appearing in patients with cancer implies an increased risk of VTE recurrences, bleeding, and mortality [1,2,3,4]. Current guidelines on antithrombotic therapy, based on the results of randomized clinical trials, provide recommendations that apply to patients with all types of cancers [5,6,7]. However, cancer is a widely heterogeneous disease, and the risk of recurrences, bleeding, or death may differ according to the cancer site or other variables [8,9,10]. Several patient-, disease-, or treatment-related factors may account for these differences, including the initial VTE presentation, different cancer therapies, the presence of metastases, or the use of central venous catheters [1], but the association between the cancer histology and outcome after a VTE event has not been consistently evaluated thus far.

Lung cancer is the most common malignancy, and a leading cause of death [11]. In a recent study on 1725 patients with lung cancer and VTE, we found that their rate of VTE recurrences during anticoagulation far exceeded the rate of major bleeding [12]. However, the histology of lung cancer was not considered. A number of studies in the literature suggest that patients with adenocarcinoma (with or without VTE) may have a worse outcome than those with squamous or other types of lung cancer [1,3,13,14,15,16]. Better knowledge of the potential impact of lung cancer histology on outcome might likely help to design better therapeutic strategies for patients with lung cancer.

The RIETE (Registro Informatizado de la Enfermedad TromboEmbólica) registry is an ongoing prospective, multicenter, multinational, observational registry of consecutive patients with objectively confirmed acute VTE (ClinicalTrials.gov identifier: NCT02832245). RIETE is currently the largest database of patients with VTE in the world [17,18,19]. The aim of the current study was to assess the potential impact of lung cancer histology on the incidence rates of VTE recurrences, major bleeding, or death during the course of anticoagulation in patients with lung cancer and acute VTE.

## 2. Methods

### 2.1. Patients

Consecutive patients with acute deep vein thrombosis (DVT) or pulmonary embolism (PE) confirmed by objective tests (compression ultrasonography or contrast venography for suspected DVT; helical computed tomography (CT) scan, pulmonary angiography, or ventilation–perfusion lung scan for suspected PE) were included. Patients who were participating in a therapeutic clinical trial with a blinded treatment were excluded.

### 2.2. Inclusion Criteria

In RIETE, we started to gather data on the histology of lung cancer on January 2021. Thus, the study included patients recruited from January 2021 to April 2022. For this study, patients with biopsy-proven active lung cancer presenting with acute VTE were included. Characteristics of patients (sex, age, body weight, comorbidities, risk factors for VTE, concomitant medications), initial VTE presentation, and anticoagulant treatment (initial [first 1–3 weeks of treatment after VTE diagnosis, depending on the type of anticoagulant used] and long-term [after the initial acute phase] therapy), blood tests at baseline, and outcomes during anticoagulation, were recorded. Active lung cancer was defined as cancer diagnosed within 6 months before VTE, metastatic disease, or treatment for cancer during the previous 6 months (chemotherapy, radiotherapy, surgery, or hormonal or support therapy).

### 2.3. Follow-Up

There was no standardization of treatment and patients were managed according to the clinical practice of each participating hospital. The anticoagulant therapy (type, dose, and duration) was recorded. All patients were followed-up during anticoagulant therapy, for at least 3 months, unless earlier death occurred. During follow-up, any signs or symptoms suggesting symptomatic VTE recurrences or bleeding events were noted. Clinically suspected symptomatic VTE recurrences were investigated by corresponding objective tests: compression ultrasonography, ventilation–perfusion lung scintigraphy, helical CT pulmonary, angiography, or pulmonary angiography. VTE recurrence was defined as a DVT in a new segment, a DVT of 4 mm larger in diameter when compared with prior venous ultrasound, a new intraluminal filling defect on a CT scan, or a new ventilation–perfusion mismatch in a repeat lung scan. Bleeding was considered as major if the event was overt and required ≥2 units of blood transfusions or more, involved a critical area (intracranial, retroperitoneal, spinal, or intrapericardial), or when it was fatal. In the absence of autopsy, fatal PE was defined as any death appearing within 10 days after PE diagnosis (either the index PE or recurrent PE), in the absence of any alternative cause of death. Fatal bleeding was defined as any death occurring within 10 days of a major bleeding episode, in the absence of an alternative cause of death.

### 2.4. Statistical Analysis

Differences across subgroups were assessed using a chi-square test (two-sided) and Fisher’s exact test (two-sided) for categorical variables, and using Student’s *t*-test for variance analysis of continuous variables. Incidence rates were calculated as cumulative incidence (events/100 patient-years) and compared using the hazard ratios (HR) with corresponding 95% confidence intervals (CIs). The analyses used time-to-event methods. Risks of VTE recurrence or major bleeding were assessed with proportional hazard Cox models. Time zero was the date of the index VTE diagnosis. Patients were censored at the time of discontinuation of anticoagulation, at the time of death, or at the last date for which outcome data were available. A competing risk analysis for VTE recurrences and major bleeding with the Fine and Gray method was performed, with overall death as the competing event. Covariates entering in the adjusted model were selected by a significance level of *p* < 0.10 on univariable analysis, or by a well-known association reported in the literature. Then, a backward selection was used for the covariate selection in the multivariable model. SPSS software (version 20; SPSS, Chicago, IL, USA) and Stata 16.1 (StataCorp, College Station, TX, USA) were used for the statistical management of the data. A two-sided *p* < 0.05 was considered to be statistically significant.

## 3. Results

As of April 2022, a total of 482 patients with lung cancer and VTE had been recruited. Of these, 334 (69%) initially presented with PE (with or without concomitant DVT) and 148 (31%) presented with isolated DVT. Overall, 293 (61%) patients had adenocarcinoma, 98 (20%) patients had squamous cell carcinoma, 44 (9.1%) had small-cell carcinoma, and 47 (9.8%) were classified as other tumors, which included 9 neuroendocrine, 7 large cells (different to adenocarcinoma/squamous), 15 undifferentiated, and 16 other types. Compared to patients with adenocarcinoma, those with squamous lung cancer were older and more likely to be men, to have chronic lung disease, anemia, or thrombocytopenia, or to be using antiplatelet drugs at baseline, but were less likely to have metastases (*p* < 0.001) (Table 1). Moreover, the index VTE appeared earlier in patients with adenocarcinoma than in those with squamous cancer (median, 2 vs. 5 months after cancer diagnosis, *p* < 0.001). In 50% of patients with adenocarcinoma, the VTE appeared within the first 90 days, as compared to only 26% of patients with squamous carcinoma (*p* < 0.001). In 83 patients with adenocarcinoma (28%), the VTE was detected less than 30 days after cancer diagnosis. Patients with small-cell cancer were also less likely to have metastases than those with adenocarcinoma (*p* < 0.05), and the VTE was also detected later (median, 4 months, *p* < 0.05). Similar findings were observed in patients with other histologies. Interestingly, 89% of patients had stage III or IV tumors, with no differences between subgroups, and only 19% had an ECOG score > 2 points.

Most patients in all four subgroups (89%) received initial therapy with low-molecular-weight heparin (LMWH), with no significant differences in daily doses between subgroups (Table 2). Then, most (82%) patients maintained LMWH as long-term therapy. The median duration of anticoagulant therapy ranged between 87 and 160 days, the difference between subgroups being not statistically significant. The proportion of patients receiving chemotherapy and/or radiotherapy also was similar in all four subgroups.

During anticoagulation (median duration 104 days, IQR: 39–209 days), 9 patients developed PE recurrences, 5 had DVT recurrences, 15 suffered major bleeding (gastrointestinal 4, intracranial 4, retroperitoneal 2, other sites 5), and 218 died (fatal PE 10, fatal bleeding 2) (Table 3). All patients dying because of PE had adenocarcinoma, and all of them died within the first 10 days after the index PE. Compared to patients with adenocarcinoma, those with small-cell carcinoma had a non-statistically significant higher rate of intracranial bleeding (hazard ratio (HR): 4.11; 95%CI: 0.56–30.3). When considering patients with squamous, small-cell, or other types as a unique subgroup, patients with adenocarcinoma had a non-significantly higher rate of VTE recurrences (HR: 2.35; 95%CI: 0.66–8.42), a non-significantly lower rate of major bleeding (HR: 0.46; 95%CI: 0.16–1.30), and a similar mortality rate (HR: 1.15; 95%CI: 0.87–1.52) relative to these (Table 4).

On multivariable analysis, patients with adenocarcinoma had a lower risk of major bleeding (HR: 0.29; 95%CI: 0.09–0.95), a non-significantly higher risk of VTE recurrences (HR: 3.79; 95%CI: 0.76–18.8), and a similar mortality risk (HR: 1.02; 95%CI: 0.76–1.36) compared to those with other tumor types (Table 4). Interestingly, the cumulative rate of VTE recurrences during anticoagulation outweighed the rate of major bleeding in patients with adenocarcinoma, while the opposite was observed in the other types of lung cancer (Figure 1 and Figure 2).

## 4. Discussion

Our findings, obtained from a large series of consecutive patients with lung cancer and VTE, reveal important differences between subgroups in their clinical characteristics, time elapsed from cancer diagnosis to VTE, and outcomes during anticoagulation. Patients with adenocarcinoma (61% of the whole series) were slightly younger, less likely to be men, and the index VTE appeared earlier after the diagnosis of cancer than in patients with other histologies. During the course of anticoagulation, the rate of VTE recurrences was up to two-fold higher than the rate of major bleeding in patients with adenocarcinoma (11 vs. 6 events, respectively), while the rate of recurrences was one-third the rate of major bleeding in those with other cancer types (3 vs. 9 events, respectively). Moreover, all 10 patients dying of PE had adenocarcinoma. Thus, our findings suggest that, during anticoagulant therapy for VTE in patients with lung adenocarcinoma, physicians should be more concerned about the effectiveness than about safety of anticoagulation. On the contrary, among patients with other types of lung cancer, the most important issue is safety. Of note, patients with squamous lung cancer presented with a higher prevalence of anemia, and thrombocytopenia, and they were on antiplatelet treatment more frequently than patients with adenocarcinoma. These variables might imply an increased risk of major bleeding events [20].

Eighty-three patients with adenocarcinoma (28%) had developed the index VTE within the first 30 days after cancer diagnosis, when some patients had not started yet oncological therapy, and we hypothesize that this may have influenced the higher mortality rate in this subgroup. Anticancer therapy may harbor some prothrombotic effects, but the control of the malignancy implies a reduction in cancer-related hypercoagulability. This might likely explain the higher rate of VTE recurrences and fatal PE in patients with adenocarcinoma, since anticoagulant therapy may also need good cancer control to be effective [21,22,23]. In a retrospective study that included 727 patients with lung cancer, those with an early VTE event had worse overall survival when compared to all other patients (median 4 vs. 17 months, *p* < 0.0001), irrespective of the disease stage (HR 2.4; 95%CI 1.6–3.3) [23]. In a prior RIETE study on 1725 patients with lung cancer-associated thrombosis, we found that 50% had been diagnosed with cancer less than 3 months before the VTE, and 32% less than 30 days. Up to 32% had not received radio- or chemotherapy yet. Almost one in every six patients (16%) with fatal PE died within the first 24 h, with little time for anticoagulant treatment to become effective. Therefore, the precise identification of patients at risk would probably help the implementation of adequate preventive measures and favor early detection of VTE [12]. In the present study, we noticed that the histology could be a key characteristic to take into account. Of note, patients with adenocarcinoma were more likely to have metastases than squamous or small-cell cancer, which could explain an increased risk of developing VTE earlier.

This study has some limitations. First, because of the sample size, the events during follow-up were scarce. Thus, the capacity to find significant differences between subgroups was limited. Second, because of the observational design of RIETE registry, anticoagulant treatment may vary according to local practices, but the majority of patients were treated similarly with LMWH for initial and long-term therapy, with a small proportion of patients receiving oral anticoagulants, and no significant differences between subgroups. Third, the study was limited to the anticoagulant treatment period and cannot provide information regarding the duration of anticoagulation. Fourth, not having a central adjudication committee could entail a risk of under- or overestimation of some outcomes, such as the cause of death, but without expected differences between subgroups of the study. Fifth, we did not have information on tumor genetics which may influence the risk of thrombosis, as some oncogenic rearrangements (for example, ALK or ROS1) are more frequent in patients with adenocarcinoma [24,25]. Overall, the design of the study only allows for generating hypotheses.

## 5. Conclusions

In conclusion, the rates of VTE recurrences and major bleeding during anticoagulation in lung cancer patients with VTE appear to vary according to the histology. In patients with adenocarcinoma, the rate of VTE recurrences outweighed the rate of major bleeding. In those with other lung cancers, the opposite occurred. If these findings are externally validated, they could be the basis to start individualizing anticoagulant therapy for VTE in patients with lung cancer.

## Figures and Tables

**Figure 1 cancers-14-04127-f001:**
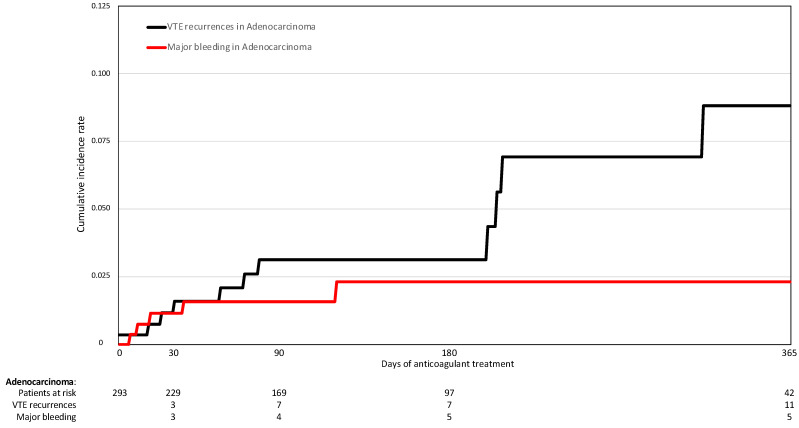
VTE recurrences and major bleeding while on anticoagulant treatment for patients with adenocarcinoma lung cancer.

**Figure 2 cancers-14-04127-f002:**
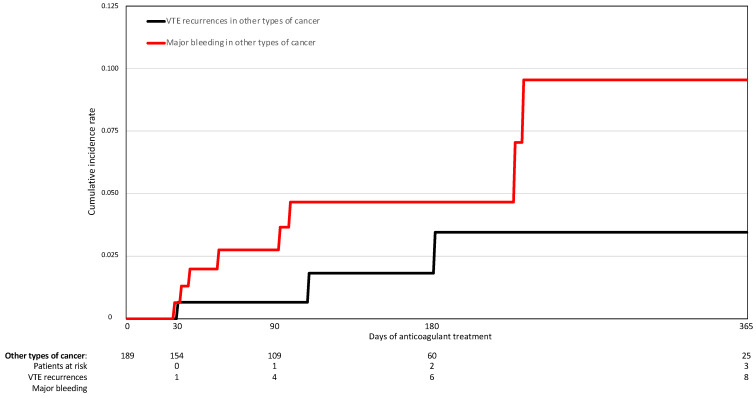
VTE recurrences and major bleeding while on anticoagulant treatment for patients with other types of lung cancer.

**Table 1 cancers-14-04127-t001:** Clinical characteristics of the patients, according to the histology.

	Adenocarcinoma	Squamous	Small-Cell	Other
** *Patients, N* **	** *293* **	** *98* **	** *44* **	** *47* **
**Clinical characteristics**				
Male gender	200 (68%)	85 (87%) ^‡^	32 (73%)	37 (79%)
Age (mean years ± SD)	63 ± 11	67 ± 9.1 ^‡^	66 ± 8.2	65 ± 8.6
Body weight (mean kg ± SD)	71 ± 13	71 ± 11	75 ± 13	72 ± 13
Smoking habit	52 (19%)	25 (27%)	11 (26%)	12 (27%)
Chronic lung disease	67 (23%)	42 (43%) ^‡^	9 (21%)	12 (26%)
Chronic heart failure	13 (4.4%)	5 (5.1%)	1 (2.3%)	2 (4.3%)
**Additional risk factors for VTE**				
Recent surgery	15 (5.1%)	6 (6.1%)	2 (4.5%)	4 (8.5%)
Immobility ≥ 4 days	64 (22%)	13 (13%)	7 (16%)	8 (17%)
Use of estrogens	5 (1.7%)	3 (3.1%)	1 (2.3%)	0
None of the above	215 (73%)	77 (79%)	35 (80%)	36 (77%)
Prior VTE	17 (5.8%)	8 (8.2%)	5 (11%)	2 (4.3%)
**Initial VTE presentation**				
Pulmonary embolism	211 (72%)	66 (67%)	32 (73%)	25 (53%) ^†^
*In patients with PE*				
SBP levels < 90 mm Hg	10 (5.1%)	1 (1.6%)	0	2 (8.3%)
Heart rate > 100 bpm (*N =* 311)	74 (38%)	20 (32%)	8 (27%)	8 (35%)
Sat O_2_ levels (mean % ± SD)	90 *±* 8.1	92 ± 4.6	93 ± 3.8	94 ± 3.7
PESI ≥ 105 points	128 (61%)	54 (82%) ^†^	22 (69%) *	17 (68%)
Lower-limb DVT	53 (18%)	20 (20%)	8 (18%)	12 (26%)
Upper-limb DVT	29 (9.9%)	14 (14%)	4 (9.1%)	10 (21%) *
**Cancer characteristics**				
Metastases	240 (82%)	53 (54%) ^‡^	30 (68%) *	31 (67%) *
Time since cancer diagnosis				
Median months (IQR)	2 (0–11)	5 (2–15) ^‡^	4 (1–11) *	3 (0–10)
<90 days	139 (50%)	24 (26%) ^‡^	13 (31%) *	21 (48%)
Stage III–IV	266 (91%)	76 (78%)	42 (96%)	43 (92%)
ECOG 3–4	61 (21%)	16 (17%)	6 (14%)	9 (19%)
**Concomitant therapies**				
Antiplatelets	43 (15%)	25 (26%) *	7 (16%)	9 (19%)
Corticosteroids	89 (30%)	24 (25%)	12 (27%)	11 (23%)
**Blood tests**				
Anemia	159 (54%)	72 (74%) ^‡^	26 (59%)	33 (70%) *
Leukocyte count > 11,000/µL	106 (37%)	29 (31%)	9 (19%)	10 (22%)
Platelet count < 100,000/µL	9 (3.1%)	10 (10%) ^†^	1 (2.3%)	3 (6.4%)
Platelet count > 450,000/µL	14 (4.8%)	8 (8.2%)	5 (11%)	9 (19%) ^‡^
CrCl levels < 60 mL/min	50 (17%)	22 (22%)	11 (25%)	8 (17%)

Comparisons among groups (Adenocarcinoma as reference group): * *p* < 0.05; † *p* < 0.01; ‡ *p* < 0.001. Abbreviations: VTE, venous thromboembolism; SD, standard deviation; CrCl, creatinine clearance; CI, confidence intervals. Anemia is defined as hemoglobin levels <12.0 g/dL in women and <13.0 g/dL in men.

**Table 2 cancers-14-04127-t002:** Treatment strategies according to the histology.

	Adenocarcinoma	Squamous	Small-Cell	Others
** *Patients, N* **	** *293* **	** *98* **	** *44* **	** *47* **
**Duration of therapy**				
Mean days (±SD)	184 ± 212	195 ± 261	214 ± 203	175 ± 236
Median days (IQR)	114 (47–210)	105 (38–218)	160 (91–232)	87 (44–203) *
Duration > 6 months	100 (35%)	30 (32%)	18 (41%)	12 (26%)
**Initial therapy**				
Low-molecular-weight heparin	260 (89%)	89 (91%)	40 (91%)	39 (83%)
Mean LMWH dose (IU/kg/day)	169 ± 41	170 ± 44	164 ± 42	164 ± 43
Unfractionated heparin	3 (1.0%)	0	0	2 (4.3%)
Thrombolytics	6 (2.0%)	0	1 (2.3%)	0
Direct oral anticoagulants	1 (0.3%)	0	0	0
Inferior vena cava filter	2 (0.7%)	1 (1.0%)	0	0
**Long-term therapy**				
Low-molecular-weight heparin	240 (82%)	74 (76%)	39 (89%)	40 (85%)
Mean LMWH dose (IU/kg/day)	159 ± 38	153 ± 37	155 ± 35	149 ± 47
Vitamin K antagonists	14 (4.8%)	6 (6.1%)	1 (2.3%)	1 (2.1%)
Direct oral anticoagulants	2 (0.7%)	1 (1.0%)	0	1 (2.1%)
**Oncological therapy**				
Chemotherapy	150 (52%)	55 (59%)	29 (66%)	23 (55%)
Radiotherapy	60 (21%)	22 (24%)	11 (25%)	10 (24%)
Other	12 (4.1%)	7 (7.1%)	5 (11%) *	1 (2.1%)

Comparisons among groups (Adenocarcinoma as reference group): * *p* < 0.05. Abbreviations: SD, standard deviation; LMWH, low-molecular-weight heparin; IU, international units; CI, confidence intervals.

**Table 3 cancers-14-04127-t003:** Clinical outcomes during the course of anticoagulant therapy, according to the histology.

	Adenocarcinoma		Squamous		Small-Cell		Other	
	*N*	Events per 100 Patient-Years	*N*	Events per 100 Patient-Years	*N*	Events per 100 Patient-Years	*N*	Events per 100 Patient-Years
** *Patients, N* **	** *293* **		** *98* **		** *44* **		** *47* **	
Patient-years of therapy	106		101		153		85	
PE recurrences	7	4.93 (1.98–10.1)	0	-	1	4.01 (0.05–22.3)	1	47.7 (0.06–26.5)
DVT recurrences	4	2.82 (0.76–7.21)	1	2.08 (0.03–11.5)	0	-	0	-
VTE recurrences	11	7.74 (3.86–13.9)	1	2.08 (0.03–11.5)	1	4.01 (0.05–22.3)	1	4.76 (0.06–26.5)
Major bleeding	6	4.22 (1.54–9.19)	4	8.30 (2.23–21.3)	3	12.0 (2.42–35.2)	2	9.53 (1.07–34.4)
Gastrointestinal	2	1.41 (0.16–5.08)	2	4.15 (0.47–15.0)	0	-	0	-
Intracranial	2	1.41 (0.16–5.08)	0	-	2	8.03 (0.90–29.0) *	0	-
Menorrhagia	1	0.70 (0.01–3.92)	0	-	0	-	0	-
Retroperitoneal	1	0.70 (0.01–3.92)	0	-	1	4.01 (0.05–22.3)	0	-
Hemopericardium	0	-	1	2.08 (0.03–11.5)	0	-	1	4.76 (0.06–26.5)
Hemoptysis	0	-	1	2.08 (0.03–11.5)	0	-	1	4.76 (0.06–26.5)
All-cause death	141	99.3 (83.6–117)	39	81.0 (57.6–110)	16	64.2 (36.7–104)	22	105 (65.7–159)
Fatal PE	10	7.04 (3.37–12.9)	0	-	0	-	0	-
Fatal bleeding	1	0.70 (0.01–3.92)	0	-	1	4.01 (0.05–22.3)	0	-

Comparisons among groups (Adenocarcinoma as reference group): * *p* < 0.05. Abbreviations: VTE, venous thromboembolism; PE, pulmonary embolism; DVT, deep vein thrombosis; IQR, interquartile range.

**Table 4 cancers-14-04127-t004:** Univariable and multivariable analyses for adverse events during the course of anticoagulation, according to cancer histology (adenocarcinoma vs. other).

	Hazard Ratio (95%CI)	*p* Value
** *Univariable analysis* **		
VTE recurrences	2.35 (0.66–8.42)	0.190
Major bleeding	0.46 (0.16–1.30)	0.145
Overall death	1.15 (0.87–1.52)	0.322
Fatal PE	6.5 (0.83–50.6)	0.075
Fatal bleeding	0.64 (0.04–10.2)	0.752
** *Multivariable analysis* **		
VTE recurrences	3.79 (0.76–18.8)	0.103
Major bleeding	0.29 (0.09–0.95)	0.042
Overall death	1.02 (0.76–1.36)	0.902
Fatal PE	5.74 (0.73–45.0)	0.096
Fatal bleeding	0.62 (0.04–9.94)	0.736

Abbreviations: VTE, venous thromboembolism; PE, pulmonary embolism; CI, confidence intervals. Multivariate analysis is a competing risk analysis for VTE recurrences and major bleeding, with overall death as the competing event. For overall death, a multivariate Cox analysis was conducted. Variables included in the multivariate analysis: anemia, initial VTE presentation (PE vs. isolated DVT), and time since cancer diagnosis <30 days, for VTE recurrences; stage IV (vs. I–III), for major bleedings; age ≥65 years, stage IV (vs. I–III), recent immobility ≥4 days, leukocyte count >11,000/µL, and ECOG 3–4 (vs. ≤2), for overall death; recent immobility ≥4 days, for fatal PE; and chronic heart failure for fatal bleeding.

## Data Availability

The data presented in this study are available on request from the corresponding author.

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
