# Peer review of "Cancer Histology and Natural History of Patients with Lung Cancer and Venous Thromboembolism"

_cancers, 2022, doi:10.3390/cancers14174127_

Round 1

Reviewer 1 Report

The authors submitted this interesting and important paper. However, I have significant concerns about the manuscript.

First concern is the grammar and syntax of the manuscript. Authors would benefit from English language editing, though these errors do not interfere with the comprehension of the manuscript. Examples of this include in the Results section of the abstract the authors should use less colloquial language (ex "and a similar risk to die" should be changed to "have a similar risk of mortality"). 

I would suggest the authors not use definitive language in the introduction, "but no studies had yet evaluated the association between cancer histology and outcome after a VTE event." There have been studies, maybe not as large as this one. Thus would soften the language in this instance.

For their analysis, I am concerned with the amount of variables included in the multi-variable analysis. For example, there are 14 variables used in the models and only 11 VTE recurrences and 6 major bleeding events. This leads to over-fitting of the model. A rule of thumb is one co-variable per 7 to 10 events. One suggestion is for composite outcomes in order to be able to perform multi-variable analysis. 

Additionally, no information on tumor genetics is a limitation that should be noted in the discussion given recent data on genetics and thrombosis risk as well as certain genetic mutations being more commonly found in adenocarcinoma compared with other histologic types (https://www.mdpi.com/2072-6694/12/7/1958). 

Author Response

Response to the reviews: cancers-1814994

Cancer histology and natural history of patients with lung cancer and venous thromboembolism

Reviewer: 1

  1. The authors submitted this interesting and important paper. However, I have significant concerns about the manuscript.

Response 1: Thank you for your interest in this submission.

  1. First concern is the grammar and syntax of the manuscript. Authors would benefit from English language editing, though these errors do not interfere with the comprehension of the manuscript. Examples of this include in the Results section of the abstract the authors should use less colloquial language (ex "and a similar risk to die" should be changed to "have a similar risk of mortality").

Response 2: We appreciate this comment from the Reviewer. We have made modifications in order to improve the grammar and syntax of the manuscript.

  1. I would suggest the authors not use definitive language in the introduction, "but no studies had yet evaluated the association between cancer histology and outcome after a VTE event." There have been studies, maybe not as large as this one. Thus would soften the language in this instance.

Response 3: Thank you for the comment. According to the Reviewer’s suggestion, we have modified the introduction.

  1. For their analysis, I am concerned with the amount of variables included in the multi-variable analysis. For example, there are 14 variables used in the models and only 11 VTE recurrences and 6 major bleeding events. This leads to over-fitting of the model. A rule of thumb is one co-variable per 7 to 10 events. One suggestion is for composite outcomes in order to be able to perform multi-variable analysis.

Response 4: We appreciate the Reviewer’s input. We agree that the low number of events is a limitation for the multivariable analysis and that a composite outcome would minimize this issue. But on the other hand, our results suggest a different behavior of the risks of VTE recurrence and bleeding, respectively, depending on the tumor histology. If these outcomes were combined the differences would be unnoticeable.

  1. Additionally, no information on tumor genetics is a limitation that should be noted in the discussion given recent data on genetics and thrombosis risk as well as certain genetic mutations being more commonly found in adenocarcinoma compared with other histologic types (https://www.mdpi.com/2072-6694/12/7/1958).

Response 5: Thank you for the comment. According to the Reviewer’s comment, we have added comments in the discussion.

Reviewer 2 Report

The authors included 482 consecutive patients with lung cancer and VTE from January 2021 to April 2022 as part of the RIETE multi-centre study. They investigated incidence of VTE recurrence, major bleeding and death during anticoagulant therapy between different histological lung cancer subtypes (adenocarcinoma, squamous cell carcinoma, small cell lung cancer and other subtypes) and found that patients with adenocarcinoma had higher and earlier incidence of VTE recurrence but suffered from major bleeding less often than patients with other subtypes. The adenocarcinoma patients were younger and less frequently male, but they had more advanced disease, meaning that age and gender could not explain higher VTE recurrence rate, but more advanced disease could possibly contribute. The results are of interest due to the perspectives for more individualized VTE and bleeding risk assessment in lung cancer patients, and may form the basis for future, larger studies designed to investigate the effect of histological subtype on VTE and bleeding risk in these patients. Below are my comments and questions for the authors.

1) The RIETE study is a multi-centre study. From which hospitals/centres were the patients in the present study included? Were the proportion of different histological subtypes similar between the different study centres? If not, this could introduce some bias in how VTE and bleeding was registered/diagnosed.

2) Page 3, lines 10-11: “During each visit, any signs or symptoms suggesting symptomatic VTE recurrences or bleeding events were noted.” I have some questions regarding the collection of VTE and bleeding symptoms/events. Were the mentioned visits in the follow-up period study-related visits or just any visit to the hospital? How frequent were those visits? Were the patients “interviewed” for VTE symptoms and bleeding symptoms by one of the study conductors, or were they interviewed by clinical personnel as part of routine check-up and info was later collected by study conductors? Finally, did you go through the patients’ medical records at the end of follow-up to identify any hospital admissions for VTE recurrence or bleeding which may not have been covered by the above-mentioned visits?

3) Page 3, line 48 to page 4, line 3: “Moreover, the index VTE appeared earlier in patients with adenocarcinoma than in those with squamous cancer (median, 2 vs. 5 months after cancer diagnosis). In 50% of patients with adenocarcinoma the VTE appeared within the first 90 days, as compared to only 26% of patients with squamous carcinoma.” Could this be because patients with adenocarcinoma were diagnosed later/had more progression of their disease than patients with squamous carcinoma or other types? In Table 1, it is evident that patients with adenocarcinoma more often had metastatic disease and higher staging. This should be mentioned in the discussion.

4) Median follow up times/number of days in the four different groups are not quite clear and should be specified in Table 1.

5) In Table 3, it is unclear what “Median days” refer to. One could assume that it is median days with anticoagulation, i.e. median days of follow-up (since patients were censored if they discontinued anticoagulation), but the numbers do not fit with table 2 where duration of anticoagulation is stated?

6) Related to my question 4, how many patients discontinued anticoagulation?

7) Did the patients with fatal PE have other VTE risk factors (e.g. surgery, immobilization)?

8) More patients with squamous carcinoma received antiplatelet therapy. This may have influenced bleeding rates.

7) A minor detail: Page 8, line 34: “Forth” should be “Fourth”.

Author Response

Reviewer: 2

  1. The authors included 482 consecutive patients with lung cancer and VTE from January 2021 to April 2022 as part of the RIETE multi-centre study. They investigated incidence of VTE recurrence, major bleeding and death during anticoagulant therapy between different histological lung cancer subtypes (adenocarcinoma, squamous cell carcinoma, small cell lung cancer and other subtypes) and found that patients with adenocarcinoma had higher and earlier incidence of VTE recurrence but suffered from major bleeding less often than patients with other subtypes. The adenocarcinoma patients were younger and less frequently male, but they had more advanced disease, meaning that age and gender could not explain higher VTE recurrence rate, but more advanced disease could possibly contribute. The results are of interest due to the perspectives for more individualized VTE and bleeding risk assessment in lung cancer patients, and may form the basis for future, larger studies designed to investigate the effect of histological subtype on VTE and bleeding risk in these patients. Below are my comments and questions for the authors.

Response 1: Thank you for your interest in this submission and the thoughtful summary.

  1. The RIETE study is a multi-centre study. From which hospitals/centres were the patients in the present study included? Were the proportion of different histological subtypes similar between the different study centres? If not, this could introduce some bias in how VTE and bleeding was registered/diagnosed.

Response 2: Thank you for the comment. The proportion of the different histologies assessed was similar among the recruiting centres.

  1. Page 3, lines 10-11: “During each visit, any signs or symptoms suggesting symptomatic VTE recurrences or bleeding events were noted.” I have some questions regarding the collection of VTE and bleeding symptoms/events. Were the mentioned visits in the follow-up period study-related visits or just any visit to the hospital? How frequent were those visits? Were the patients “interviewed” for VTE symptoms and bleeding symptoms by one of the study conductors, or were they interviewed by clinical personnel as part of routine check-up and info was later collected by study conductors? Finally, did you go through the patients’ medical records at the end of follow-up to identify any hospital admissions for VTE recurrence or bleeding which may not have been covered by the above-mentioned visits?.

Response 3: Thank you for the comment. As Riete registry is an observational prospective cohort study there was no standardization of follow-up visits or treatment and patients were managed according to the clinical practice of each participating hospital. Thus, the events during the follow-up are collected during routine check-up by responsible physician of patients. However, the responsible physician usually is the principle investigator of Riete registry in each center. Patients are followed up for a minimum of 3 months and usually every 3 months during anticoagulation. The follow-up finishes when the last visit is performed. We have added some comments to clarify this question in Methods section.

  1. Page 3, line 48 to page 4, line 3: “Moreover, the index VTE appeared earlier in patients with adenocarcinoma than in those with squamous cancer (median, 2 vs. 5 months after cancer diagnosis). In 50% of patients with adenocarcinoma the VTE appeared within the first 90 days, as compared to only 26% of patients with squamous carcinoma.” Could this be because patients with adenocarcinoma were diagnosed later/had more progression of their disease than patients with squamous carcinoma or other types? In Table 1, it is evident that patients with adenocarcinoma more often had metastatic disease and higher staging. This should be mentioned in the discussion.

Response 4: Thank you for the excellent suggestion. We have added comments in the discussion.

  1. Median follow up times/number of days in the four different groups are not quite clear and should be specified in Table 1.

Response 5: Thank you for the comment. The median follow-up in the four different groups is indicated in Table 2, which is the duration of anticoagulant therapy (mean and median), because the follow-up considered for this study was the anticoagulation period.

  1. In Table 3, it is unclear what “Median days” refer to. One could assume that it is median days with anticoagulation, i.e. median days of follow-up (since patients were censored if they discontinued anticoagulation), but the numbers do not fit with table 2 where duration of anticoagulation is stated?

Response 6: Thank you for the excellent appreciation. Effectively, there was a mistake in Table 3 where we used time-to-event methods, so it is the number of patient-years and not the median days, as in Table 2. Corrected in the new revised version.

  1. Related to my question 4, how many patients discontinued anticoagulation?

Response 7: There were 47 patients with adenocarcinoma and 31 with other histologies, who discontinued anticoagulation for any reason (death, bleeding, discontinuation).

  1. Did the patients with fatal PE have other VTE risk factors (e.g. surgery, immobilization)?

Response 8: None of 10 patients with fatal PE had a recent surgery and 6 had immobility ≥4 days.

  1. More patients with squamous carcinoma received antiplatelet therapy. This may have influenced bleeding rates.

Response 9: Excellent appreciation! Included in the discussion.

  1. A minor detail: Page 8, line 34: “Forth” should be “Fourth”.

Response 10: Thank you. Modified accordingly.

Reviewer 3 Report

The authors utilised the RIETE registry to evaluate VTE and bleeding in patients with lung cancer of different histology types. They report on the outcome of VTE, time to VTE, and bleeding events. They demonstrate that there is a trend towards higher rates of VTE and fatal PE in patients with adenocarcinoma with less bleeding events.

Overall this is an interesting research manuscript with a large cohort to evaluate lung cancer subtypes. My main concern is regarding the authors overstating their findings, as many of the finding do not reach statistical significance. The manuscript should be rewritten to address this and can suggest trends towards VTE/fatal PE, but should not magnify these findings.

Minor comments

1.     In the introduction, the sentence “Several patient-, disease- or treatment-related factors may account for these differences, including the initial VTE presentation, different cancer therapies, the presence of metastases or the use of central venous catheters, but no studies had yet evaluated the association between the cancer histology and outcome after a VTE event” requires referencing for the factors mentioned

Methods

-       Can the authors define what is meant by initial and long term anticoagulation

Results

-       The authors mention that 9.8% of patients have ‘other tumours’ – what does this include? If this is a heterogenous group, should it be excluded?

-       Suggest rewording the sentence ‘Overall, 293 patients had adenocarcinoma (61%), squamous 98 (20%), small cell 44 (9.1%), other tumors 47 (9.8%)’. Consider ‘Overall, 293 patients had an adenocarcinoma, 98 (20%) patients had a squamous cell carcinoma, 44 (9.1%) had a small cell carcinoma, and 47 (9.8%) were classed as other tumours, which included…’. Also, the percentages only add up to 99.9%

-       There are several statistics in the results (timing of index VTE, VTE in first 90 days etc). The authors should provide measures of significance (p values or odds ratio for these)

-       ‘Compared to patients with adenocarcinoma, those with small cell carcinoma had a higher rate of intracranial bleeding (hazard ratio [HR]: 4.11; 95%CI: 0.56-30.3)’ – This finding is non-significant and should be stated.

-        

Discussion

Although the authors findings demonstrate a trend towards significance with higher recurrences and higher fatal PE in the adenocarcinoma group, the findings are non-significant. Suggest that the discussion regarding effectiveness/safety is toned down (first paragraph), and similarly in the conclusion. Although these findings are interesting, they do not reach statistical significance and cannot be overstated.

Author Response

Reviewer: 3

  1. The authors utilised the RIETE registry to evaluate VTE and bleeding in patients with lung cancer of different histology types. They report on the outcome of VTE, time to VTE, and bleeding events. They demonstrate that there is a trend towards higher rates of VTE and fatal PE in patients with adenocarcinoma with less bleeding events.

Overall this is an interesting research manuscript with a large cohort to evaluate lung cancer subtypes. My main concern is regarding the authors overstating their findings, as many of the finding do not reach statistical significance. The manuscript should be rewritten to address this and can suggest trends towards VTE/fatal PE, but should not magnify these findings.

Response 1: Thank you for your interest in this submission and the thoughtful summary.

  1. In the introduction, the sentence “Several patient-, disease- or treatment-related factors may account for these differences, including the initial VTE presentation, different cancer therapies, the presence of metastases or the use of central venous catheters, but no studies had yet evaluated the association between the cancer histology and outcome after a VTE event” requires referencing for the factors mentioned.

Response 2: Thank you for the comment. Done.

  1. Can the authors define what is meant by initial and long term anticoagulation.

Response 3: We have added the definitions in the Methods section. Thank you for the suggestion.

  1. The authors mention that 9.8% of patients have ‘other tumours’ – what does this include? If this is a heterogenous group, should it be excluded?

Response 4: There were 9 neuroendocrine, 7 large cells (different to adeno/squamous), 15 undifferentiated, and 16 others (specific type not registered). We have added this information in the revised manuscript.

  1. Suggest rewording the sentence ‘Overall, 293 patients had adenocarcinoma (61%), squamous 98 (20%), small cell 44 (9.1%), other tumors 47 (9.8%)’. Consider ‘Overall, 293 patients had an adenocarcinoma, 98 (20%) patients had a squamous cell carcinoma, 44 (9.1%) had a small cell carcinoma, and 47 (9.8%) were classed as other tumours, which included…’. Also, the percentages only add up to 99.9%.

Response 5: Thank you for the suggestion. Done.

  1. There are several statistics in the results (timing of index VTE, VTE in first 90 days etc). The authors should provide measures of significance (p values or odds ratio for these).

Response 6: Thank you for the comment. Done.

  1. ‘Compared to patients with adenocarcinoma, those with small cell carcinoma had a higher rate of intracranial bleeding (hazard ratio [HR]: 4.11; 95%CI: 0.56-30.3)’ – This finding is non-significant and should be stated.

Response 7: Excellent appreciation! Modified accordingly.

  1. Although the authors findings demonstrate a trend towards significance with higher recurrences and higher fatal PE in the adenocarcinoma group, the findings are non-significant. Suggest that the discussion regarding effectiveness/safety is toned down (first paragraph), and similarly in the conclusion. Although these findings are interesting, they do not reach statistical significance and cannot be overstated.

Response 8: Thank you for the comment. According to the Reviewer’s comment, we have modified the discussion and conclusion.

Round 2

Reviewer 1 Report

The authors have made some of the changes suggested and the manuscript has improved English grammar though with some small typos and grammar errors that need to be improved.

However, the authors failed to address my biggest concern with this paper. That concern is using multi-variable models with almost more variables than outcomes, leading to over-fitting of the models. I agree with the concern of composite outcomes diluting any differences between groups, however with low number of events it is inappropriate to include a large number of co-variables into the models (rule of thumb is using one co-variable for every 7 to 10 outcomes). I recommend a statistical consultation. 

Author Response

  1. The authors have made some of the changes suggested and the manuscript has improved English grammar though with some small typos and grammar errors that need to be improved.

Response 1: Thank you for the suggestion. We have made more modifications in order to improve significantly the grammar of the manuscript.

  1. However, the authors failed to address my biggest concern with this paper. That concern is using multi-variable models with almost more variables than outcomes, leading to over-fitting of the models. I agree with the concern of composite outcomes diluting any differences between groups, however with low number of events it is inappropriate to include a large number of co-variables into the models (rule of thumb is using one co-variable for every 7 to 10 outcomes). I recommend a statistical consultation.

Response 2: We really appreciate this comment from the Reviewer because it is a very important point for the study and maybe, it was not clear in the manuscript. We fully agree with the Reviewer on the limitation that we have a low number of events, which makes the multivariable analysis difficult. The covariates indicated in the manuscript were evaluated to adjust the multivariable analysis, but not all were included in the final model for each type of event. We have consulted with the statistician and he confirms, in the line of the Reviewer, that there may be an overfitting, but not many variables were finally included in the multivariate models. In any case, this is an exploratory analysis and we have modified the manuscript in order to clarify this important point commented by the Reviewer. Covariates retained in the final multivariate model were:

VTE recurrences

Anemia, PE index event (vs DVT), time since cancer diagnosis <30 days

Major bleeding

Stage IV

Overall death

Age ≥65yr, stage IV, immobility ≥4 days, leukocyte > 11,000/µL, ECOG 3-4 (vs ≤2)

Fatal PE

Immobility ≥4 days

Fatal bleeding

Chronic heart failure

Round 3

Reviewer 1 Report

The authors have made improvements in their manuscript and have obtained statistical consultations. The multi-variable models have the appropriate number of co-variables.

I recommend a few minor edits prior to acceptance:

1. In Table 4 would update the variables included in the multivariable analysis since they were changed. 

2. In the discussion at the end of the first paragraph: Is there an explanation of why patients with adenocarcinoma have increased anemia, thrombocytopenia and antiplatelet therapies (increased cardiovascular co-morbidities including coronary artery disease, chemotherapy types, etc?)

3. In the discussion (third paragraph), since you have limited the number of co-variables in the multivariable models, you do not have to state there was overfitting.
